# Vitamin K Effects on Gas6 and Soluble Axl Receptors in Intensive Care Patients: An Observational Screening Study

**DOI:** 10.3390/nu13114101

**Published:** 2021-11-16

**Authors:** Ulf Schött, Cecilia Augustsson, Luukas Lilover, Caroline Ulfsdotter Nilsson, Louise Walther-Sturesson, Thomas Kander

**Affiliations:** 1Anaesthesia & Intensive Care, Department of Clinical Sciences, Lund University, Skane University Hospital, 22185 Lund, Sweden; caroline.nilsson@med.lu.se (C.U.N.); louise.walther-sturesson@med.lu.se (L.W.-S.); thomas.kander@skane.se (T.K.); 2Division of Laboratory Medicine, Department of Clinical Chemistry and Pharmacology, Skåne University Hospital, 20502 Malmö, Sweden; Cecilia.Augustsson@skane.se; 3Division of Anaesthesia and Intensive Care, Department of Clinical Sciences, Lund University, 22185 Lund, Sweden; luukas.lilover@gmail.com

**Keywords:** Gas6, Axl receptor, endothelial dysfunction, prothrombin time, Gla protein, thrombin generation, intensive care, vitamin K

## Abstract

Growth arrest-specific gene 6 protein (Gas6) is avitamin K-dependent tissue bound protein. Gas6 has been shown to promote growth and therapy resistance among different types of cancer as well as thromboembolism. The aim of this prospective screening study: ClinicalTrials.gov; Identifier: NTC3782025, was to evaluate the effects of intravenously administered vitamin K1 on Gas6 and its soluble (s)Axl receptor plasma levels in intensive care patients. Vitamin K1 was intravenously injected in non-warfarin treated patients with prolonged Owren prothrombin time international normalized ratio (PT-INR) > 1.2 and blood samples were retrieved before and 20–28 h after injection. Citrate plasma samples from 52 intensive care patients were analysed for different vitamin K dependent proteins. There was a significant, but small increase in median Gas6. Only one patient had a large increase in sAxl, but overall, no significant changes in sAxl Gas6 did not correlate to PT-INR, thrombin generation assay, coagulation factors II, VII, IX and X, but to protein S and decarboxylated matrix Gla protein (dp-ucMGP). In conclusion, there was a small increase in Gas6 over 20–28 h. The pathophysiology and clinical importance of this remains to be investigated. To verify a true vitamin K effect, improvement of Gas6 carboxylation defects needs to be studied.

## 1. Introduction

Vitamin K plays an important part in maintaining haemostasis, and its effect on coagulation was noted in experiments as far back as 1929. Vitamin K is necessary for the carboxylation of coagulation factors (F) FII, FVII, FIX, FX and proteins C, S, Z and M [1], first then they can bind Ca^2+^ and interact with phospholipid cell surfaces. The benefits of vitamin K have expanded beyond the subject of haemostasis [1,2]. Extensive research is being carried out on the different extrahepatic vitamin K-dependent (γ-carboxyglutamic (Gla)) proteins, such as osteocalcin (OC), matrix Gla protein (MGP), periostin, transthyretin, inter-alpha inhibitor heavy chain H2 and proline-rich Gla proteins 1–4. Most of the current research on extrahepatic vitamin K-dependent Gla proteins and vitamin K supplementation has focused on OC effects on bone metabolism [3] and MGP inhibition of arterial calcification [4,5]. 

Another extrahepatic vitamin K–dependent protein is the growth arrest-specific gene 6 (Gas6) protein. Gas6 has a molecular structure close to protein S, but with a much lower plasma concentration. Whereas protein S has important anticoagulative effects, these are lacking in Gas6. Conversely, high Gas6 plasma concentrations have been shown in patients with recurrent venous thromboembolism (VTE) [6]. Gas6 has been found to play a role in platelet aggregation in animal models, with Gas6-null mice showing resistance to thrombosis and embolism. Treatment with inhibition of Gas6 binding to the Tyro3, Axl and MerTK (TAM) (see below) family of receptors may provide a future treatment model against pulmonary embolisms and arterial and venous thromboses [7]. 

Gas6 is a ligand to the TAM family of receptors, a subfamily of receptor tyrosine kinases (RTKs), through which it can activate the anti-apoptotic protein kinase B as well as mitogen-activated protein kinases. Gas6 has 100–1000 times higher affinity for Axl than the other TAM receptors [8]. Gas6 and Axl has been shown to promote growth and therapy resistance among different types of cancer. Axl also has Gas6 independent effects and can be activated through protein S and other ligands [9]. 

Whereas vitamin K intake, primarily vitamin K2 (menaquinones), is mostly associated with anti-cancer effects [10], the tumour- and thrombosis-promoting properties of Gas6 raise a concern as to whether vitamin K can be safely used in patients with cancer and increased risk for thromboembolism, many of whom have signs of vitamin K deficiency with increased plasma markers for decarboxylated Gla proteins [2,10]. Gas6 needs vitamin K for its carboxylation to become a functional Gla protein. Gas6 has a mitogenic effect on metastatic human prostatic cancer cell lines expressing the Axl receptor [11]. Gas6-Axl signalling leads to increased cell survival under conditions that do not allow cell proliferation [12]. In patients with non-small cell lung cancer brain metastases, both Gas6 and Axl expression predicted worsened survival [13]. However, this is a complex field and other tumours can be inhibited by an increased Gas6 and TAM expression [8]. 

Vitamin K and Gas6 play important roles in the human nervous system as well. Vitamin K is closely linked with sphingolipid synthesis, while Gas6 via receptor signalling is involved in chemotaxis, mitogenesis, cell growth, myelinogenesis and anti-apoptosis [14]. Interestingly, Gas6 is also important for microglial processes in the brain. TAM receptors proved important for microglia in both non-inflammatory clearance of apoptotic cells and in microglial convergence on sites of injury [15]. 

Gas6 has also a vascular protective role in synergy with MGP. MGP prevents vascular calcification and Gas-6 affects vascular smooth muscle cell apoptosis and movement [16]. In atherosclerotic plaques Axl is downregulated, but MerTK receptors from invasive macrophages are upregulated and interact with protein S [17]. 

Gas6 has also been studied in intensive care. Gas6 is shown to attenuate the effects of acute lung injury induced by ischemia-reperfusion (IR) by limiting IR-related lung oedemas, inflammation and lung tissue damage through an anti-inflammatory pathway mediated by suppressor of cytokine signalling 3 (SOCS3) and upregulated via phosphorylation of Axl [18]. However, Otulakowski et al. showed that mechanical ventilation stretched open calcium channels and reduced Axl receptors’ sensitivity towards Gas6 [19]. Several studies confirm an elevated concentration of Gas6 in patients with sepsis, often with a strong correlation with the number of failing organs or organ damage score [12,20,21]. Most sepsis studies have failed to correlate Gas6 with mortality, whereas a prospective cohort study did [22]. The relevance of the increased Gas6 is unclear, is it an acute phase reactant or is the increase an anti-inflammatory, antiapoptotic response trying to increase cell survival? An example of the latter is the Gas6-Axl protection of endothelial tight junctions in sepsis [23]. 

Vitamin K treatment in hospitalized patients has mainly been focused on reversal of warfarin. However, many patients are treated with intravenous vitamin K1 on suspicion of vitamin K deficiency, with or without prolonged prothrombin times (PTs) [24]. In a recent publication, intravenous vitamin K1 at a high dose of 10 mg improved, but did not fully normalise, both vitamin K-dependent coagulation and carboxylation of extrahepatic Gla proteins after 24 h [25]. Perhaps repeated doses are better [26], but a review in the New England Journal of Medicine (NEJM) recommended intravenous vitamin K1 10 mg/week in hospitalized patients [27]. Outside hospitals, vitamin K research has mainly focused on optimal nutritional intake of vitamins K1 and K2 and vitamin K2 supplementation [28]. In vivo effects of intravenous vitamin K1 on Gas6 and Axl has not been studied in patients before. 

The aim of this prospective screening study was to evaluate the levels of Gas6 and its soluble Axl receptor before and after administration of vitamin K1 in intensive care patients with prolonged PTs and to correlate those plasma levels and their delta changes with other vitamin K–dependent proteins. Thrombin generation assay was used to evaluate overall coagulation activity.

## 2. Materials and Methods

### 2.1. Study Design

This study was a substudy of a previously published main study in *Nutrients* [25], conducted at the general intensive care unit (ICU) at Skåne University Hospital, Lund, Sweden. The Regional Ethics Board in Lund approved the study (20 December 2018; DNR 2018/1010) as well as an approved biobank (DNR 136, VO IPV, SUS, Lund, Sweden). Written informed consent was collected from all patients or closest relative as required by the board. The study was also registered at ClinicalTrials.gov (identifier NTC3782025). All patients were treated according to the standard routines at the ICU. The inclusion criteria for participation in the study were patients aged 18 or older with elevated routine Owren PT-international normalised ratio (PT-INR > 1.2) and prescribed 10 mg intravenous vitamin K1 (phytomenadione Konakion Novum^®^, Cheplapharm Arzneimittel GmbH, Greifswald, Germany) by their treating ICU doctor. The exclusion criteria were patients on warfarin or novel oral anticoagulants, who had been prescribed prothrombin complex concentrate and plasma, or who had already been treated with Konakion in the previous 36 h. Patients with known hereditary coagulative disorders and those with hepatocellular carcinoma or liver resection within the previous 6 months were also excluded from the study. Patients were also categorised according to sequential organ failure assessment (SOFA) scores. SOFA is a well-established daily score for the number of failing organ systems in critically ill patients [24].

### 2.2. Blood Sampling and Laboratory Analyses

Samples were taken ahead of vitamin K1 treatment and then again after 24 h (range 20–28 h). This interval is routinely used for warfarin reversal with vitamin K1. A previous register study indicated an improvement of PT-INR after 24 h with intravenous vitamin K1 to intensive care patients not on warfarin [24]. The blood samples were retrieved with an indwelling radial arterial catheter with a continuous flushing system and a sampling membrane, centrifuged and frozen as previously described [25]. Analyses of PT Owren, PT Quick, specific coagulation factors II, VII, IX, X, protein C, protein S, thrombin generation assay (TGA), desphospho-carboxylated MGP (dp-ucMGP) and C-reactive protein (CRP) were performed as previously described [25].

### 2.3. Enzyme-Linked Immunosorbent Assay (ELISA) for the Quantification of Growth Arrest–Specific Gene 6 (Gas6) in Human Plasma

Citrate plasma samples were analysed for Gas6 using a Gas6 ELISA and a duoset ELISA system from R&D Systems (cat. no. DY885B). Plates were washed after all incubation steps with PBST (phosphate-buffered saline with 0.1% Tween20). An assay volume of 100 µL was used. Plates (Maxisorp, Nunc) were coated with coating antibody (R&D Systems, DY885B) diluted 180-fold overnight at 4 °C. After blocking (PBST with 1% bovine serum albumin (BSA)) for 4 h at room temperature, samples were added in 10-fold dilution into sample buffer (blocking buffer with 10 mM EDTA) and incubated overnight at 4 °C. Secondary antibody was added diluted 180-fold and incubated for 3 h at room temperature, and then streptavidin-HRP was added for 20 min at room temperature. Subsequently, tetramethylbenzidin (TMB, Sigma-Aldrich) was used as a substrate, and the reaction was stopped with 0.5 M H_2_SO_4_. The absorbance was measured at 450 nm with a microplate reader. All the tests were performed at Malmö Coagulation Laboratory. A plasma pool of 50 individual normal donor plasma was used as a source of reference plasma and was measured to contain a Gas6 level of 14.9 ng/mL, with an inter-CV of 6.2%.

### 2.4. ELISA Assay for the Quantification of Soluble Axl (sAxl) Receptors

Citrate plasma samples were analysed for soluble Axl using an Axl ELISA and a duoset ELISA system from R&D Systems (cat no. DY154). Plates were washed after all incubation steps with PBST (PBS with 0.1% Tween20). An assay volume of 100 µL was used. Plates (Maxisorp, Nunc) were coated with coating antibody diluted 180-fold (R&D Systems, 841007) overnight at 4 °C. After blocking (PBST with 1% BSA) for 4 h at room temperature, samples were added in 20-fold dilution into sample buffer (blocking buffer with 10 mM EDTA) and incubated overnight at 4 °C. Secondary antibody was added diluted 180-fold and incubated for 3 h at room temperature, and then streptavidin-horse radish peroxidase (HRP) was added for 20 min at room temperature. Subsequently, TMB (Sigma-Aldrich) was used as a substrate, and the reaction was stopped with 0.5 M H_2_SO_4_. The absorbance was measured at 450 nm with a microplate reader. All the tests were performed at Malmö Coagulation Laboratory. A plasma pool of 50 individual normal donor plasma was used as a source of reference plasma and was measured to contain an sAxl level of 13.5 ng/mL, with an inter-CV of 17.6%.

### 2.5. Statistical Analyses

For the statistical analyses, GraphPad Prism (GraphPad software, La Jolla, CA, USA) was utilised to do a two-tailed Wilcoxon matched-pairs signed rank test that was in turn used to estimate changes in the before and after levels of Gas6 and sAxl receptor. Spearman correlation analyses were used to evaluate correlations. All variables were found to have non-Gaussian distributions.

## 3. Results

### 3.1. Patient Population

Blood samples before and 24 h (range 20–28 h) after intravenous vitamin K1 from 52 patients with prolonged PT-INR (1.3–1.8) were included. If not otherwise stated *n* = 52 in all paragraphs and figures. The studied cohort has previously been described [25]. In summary 69% were male, the median age was 68 years (range: 55–74 years) and septic shock was the most common diagnosis (29%) followed by cardiovascular disease 13%). Three patients (6%) were diagnosed with cancer.

### 3.2. Gas6 and sAxl Plasma Concentrations before and after Vitamin K

There was a significant increase in Gas6 (*p* < 0.01) (Figure 1a). There were no significant changes in the sAxl receptor (Figure 1b).

### 3.3. Subgroup Analyses

To better understand if changes in Gas6 and sAxl receptor could be explained by improvement or deterioration in patient condition over time, the cohort was split into groups depending on whether SOFA score or CRP increased, or remained unchanged (patient condition worsened) or decreased (patient condition improved). Gas6 increased significantly (*p* = 0.03) both in patients with increased CRP (*n* = 29) (Figure 2a) and in patients with decreased SOFA-score (*n* = 25, *p* = 0.002) (Figure 2b). There were no significant changes in sAxl receptor when analyses were split according to SOFA and CRP changes. It should be noted that the number of observations is lower than in the main analyses which increases the risk for statistical Type 2 errors.

### 3.4. Correlations between Gas6 and Other Analyses

Positive and statistically significant correlations between Gas6 and sAxl, Gas6 and Protein S and Gas6 and dp-ucMGP were found for all analyses (before and after vitamin K1 for the 52 patients). There were no significant correlations between Gas6 and TGA with two different reagents: TGA RB, which has a low concentration of phospholipids and TF, and TGA RC High, which has a high concentration of TF. There was no significant correlation between Gas6 and protein C (Figure 3). The other vitamin K dependent procoagulative factors II, VII, IX and X did not correlate with Gas6. Neither did PT Quick or PT Owren.

### 3.5. Correlations between Delta (*Δ*) Changes

Correlations between delta (Δ) changes in Gas6 and corresponding delta changes in PT-INR and dp-ucMGP were also studied to better define changes over time from before and 24 h after vitamin K1 for the 52 patients. There were no correlations between delta Gas6 values and delta PT-INR (Owren), delta Gas6 delta and dp-ucMGP or delta Gas6 and delta CRP (Figure 4).

## 4. Discussion

The main objective of this study was to investigate the potential effects of vitamin K1 treatment on Gas6 concentrations and its soluble receptor sAxl in plasma. There was a small, but significant increase in Gas6 at 24 h, but not in sAxl after vitamin K1 treatment. The concentration of Gas6 correlated to both sAxl receptor and dp-ucMGP. A sensitivity analysis showed that Gas6 increased both in patients with improved and impaired conditions, indicating that the Gas6 increase could be a true vitamin K1 effect on its synthesis and not caused by any change in the condition of the patients between the sample occasions. The delta Gas6 did not correlate to delta CRP, possibly negating a Gas6 acute phase response.

The patients included were all given vitamin K1 according to the study protocol. Treatment with vitamin K1 is generally viewed as safe, but there is a small risk of anaphylactoid reactions with intravenous administration [25]. Evidence-based recommendations for vitamin K1 treatment in critically ill non-warfarin patients are lacking, but reflect routines from warfarin reversal [27]. Vitamin K is sometimes administered to correct prolonged PTs, of unclear clinical impact in non-bleeding patients and in patients planned for invasive procedures.

The results we obtained from our Gas6 ELISA fit rather well with the reference values provided as 13–23 ng/mL [29]. Median values from both before and after vitamin K1 treatment, from 16.1 to 17.2, remain within the reference normal value range. This indicates that our ELISA is accurate relative to earlier tests. ELISAs were also tested in plasma from a healthy donor pool, before analysing the frozen plasma vials from the present study. The increased Gas6 individual values in many of the patients corroborate those from other intensive care studies [12,20,21,22].

A significant but low correlation, (r = 0.33) was observed between Gas6 and protein S. Gas6 and protein S are very similar in structure, sharing both 43% amino-acid sequence identity and the same domain structure [30,31]. However, protein S is much more abundant in human plasma [32]. Protein S can bind to Axl, but with less affinity than for Gas6 [9]. Protein S has mainly been discussed as a cofactor to activated protein C in degrading FVIIIa and FVa, thereby being a very important coagulation inhibiting protein [32]. No significant correlation between Gas6 and protein C was found in the present study. Interestingly, there were no significant changes in proteins C or S in the main study after intravenous vitamin K1 [25], even though protein S has a very short half-life of 8 h corresponding to FVII’s half-life of 7 h. No half-life of Gas6 has been presented in humans. These facts possibly negate that the increase in Gas6 was related to a real K vitamin induced synthesis. Gas6 can be released from various tissues, also vascular and be part of an antiapoptic response.

There was no correlation between Gas6 and TGA and, furthermore, no increase in TGA after vitamin K1 treatment even if PT decreased as reported in the main study (but within normal range) [25]. One patient had a very marked increase in Gas6 (Figure 1a) not linked to a high TGA, improvement in PT-INR or dp-ucMGP (Figure 4). However, this patient had a high increase in CRP, but not in sAxl (Figure 3 and Figure 4), stressing other mechanisms than vitamin K1 increase of Gas6 synthesis. TGA has been suggested to be a test of hypo- or hypercoagulation [33]. However, the exact mechanisms behind Gas6 induced thrombo-embolism are unclear, much of the research has been performed on knockout mice [34]. The following mechanisms have been highlighted in venous thromboembolism: Gas6 activation of α_IIb_β_3_ integrin_,_ binding of platelets by P-selectin glycoprotein ligands and bridging between cells expressing phosphatidylserine and TAM receptors, Gas6 signaling via Axl induces TF expression in endothelial cells, Gas6 regulates the thrombin-induced expression of endothelial adhesion receptors VCAM-1, Gas6 regulates thrombin signaling on endothelial cells supporting CCL2-mediated recruitment of proinflammatory CCR2^hi^CX3CR1^lo^ monocytes by increasing PSGL-1 expression, thus augmenting thrombus growth [34,35]. 

Increased plasma Gas6 and possible negative effects in critically ill and cancer patients is a new research area, highlighting a possible risk with vitamin K1 treatment. There has been no study on Gas6 changes after vitamin K1 administration to critically ill or perioperative patients. Vitamin K defects in the perioperative period and in intensive care patients have previously been detected by tests that are more specific than PT, analysing carboxylation defects with proteins induced by vitamin K absence (PIVKA) and dp-ucMGP [36,37,38,39]. There was a significant, but low correlation (r = 0.19) between Gas6 and dp-ucMGP in the present study. In the main study, dp-ucMGP decreased as a true sign of improved carboxylation after vitamin K1 [25]. There are no commercial tests available for measurement of human Gas6 carboxylation that could better validate a true vitamin K1 effect in our study. In vitro experiments verify that γ-carboxylation is important not only for binding of Gas6 to phospholipid membranes, but for Gas6 effects on endothelial cell survival [40]. The effect of fully carboxylated Gas6 on various types of cancer? Uncarboxylation of Gas6 with warfarin increases pancreatic cancer cell growth and metastases [41].

Subclinical vitamin K deficit (often defined as PT Owren values within the normal range 0.9–1.2 and an increased PIVKA plasma level) is common in critically ill patients [39]. Studies on the effect of vitamin K1 on these patients are few. Then, the concern is again the counterproductive effect of a potential improvement of Gas6 activity by increased carboxylation in cancer and thromboembolism. Many cancer and critically ill patients have nutritional, malabsorption, and metabolic defects with an increased risk for vitamin K deficiency [10,24,25]. Prolonged PT values on admission correlate with mortality and morbidity in general intensive care unit patients [42]. Could a more aggressive correction with vitamin K change the outcome? This is an unresolved issue. Clinical routines regarding how to treat slightly prolonged PT values in the perioperative and intensive care periods are lacking [27]. Even further, the concept of subclinical vitamin K deficiency is not generally discussed outside malnutrition or malabsorption medicine, even though it occurs frequently in the perioperative and intensive care periods. 

Gas6 needs interaction with Axl receptors for most of its biological effects [41,43,44,45]. Gas6 and sAxl had a significant correlation of r = 0.44 in the present study. Axl receptors from the endothelium/vascular wall can be shed in several clinical situations [46,47]. One patient had a marked increase in sAxl, that could indicate shedding (Figure 1b). We did not detect any significant change in sAxl receptors after vitamin K1 treatment. Some of the individual sAxl values were high, corroborating other intensive care patient studies with sepsis and coronary patients. The Gas6-sAxl complex in plasma is inactive [20]. sAxl has a surplus plasma concentration as compared to Gas6 and functions as a sink for Gas6 released from tissues. This can have negative effects on Gas6 interactions with endothelial Axl-receptors and possibly other tissue bound Axl receptors. The Gas6-sAxl plasma concentration does not necessarily reflect Gas6-Axl tissue activities. The tissue bound Axl receptors have more relevance for tumour growth and progression, but this is still an experimental research area. Macrophage invasion in solid tumours, with their release of Gas6 is also a research area [48]. Finally, Gas6 and TAM receptors [17] are affected by genetic polymorphisms, and specific Gas6 haplotypes can protect against both stroke [49] and acute coronary events [50].

### Limitations

We recognize the limitations in the present study given its before and after design. This is partly addressed by additional analyses that describe a possible direct effect of the vitamin K1 administration, but a time dependent bias cannot be ruled out. Furthermore, the study is explorative without sample size calculation for any outcomes and hypothesis-generating and only aimed to describe any change in the Gas6 or sAxl receptor plasma concentrations 20–28 h after vitamin K1 administration and does not describe any positive or negative effects of the increased Gas6 concentration. The demand for repetitive analyses over longer intensive care or postoperative periods and repeated doses of vitamin K1 and ELISAs for detection of effects on carboxylation are needed to bring this research on.

## 5. Conclusions

A small increase in median Gas6 was seen 20–28 h after intravenous vitamin K1 administration, but there was no change in the median level of sAxl receptor. The Gas6 increase was more evident in some of the patients. The mechanisms behind these individual Gas6 increases are unclear: acute phase response, increased release from tissues and vascular walls rich in Gas6 due to endotheliopathy, antiapoptic mechanism trying to protect tissues in critical ill patients, release from inflammatory macrophages/leukocytes or a true vitamin K effect on Gas6 synthesis? In vivo vitamin K1 effects on Gas6, should involve methods to study effects on its carboxylation in future studies, as has evolved with research on other vitamin K dependent proteins. Up to now, only laboratory designed Gas6 carboxylation tests exists, used in experimental research.

## Figures and Tables

**Figure 1 nutrients-13-04101-f001:**
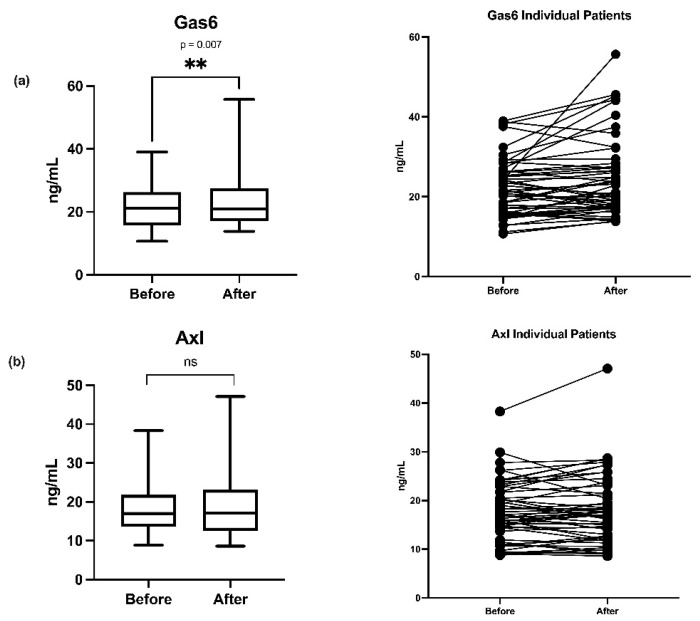
(**a**) Boxplots of growth arrest-specific gene 6 protein (Gas6) plasma concentration changes before and 24 h after vitamin K1 administration, boxplot analyses and individual patient (*n* = 52) changes. (**b**) Boxplots and individual patient (*n* = 52) plasma concentration changes of soluble Axl receptor (Axl) before and 24 h after vitamin K1. ** (*p* = 0.007) and ns (no significant change).

**Figure 2 nutrients-13-04101-f002:**
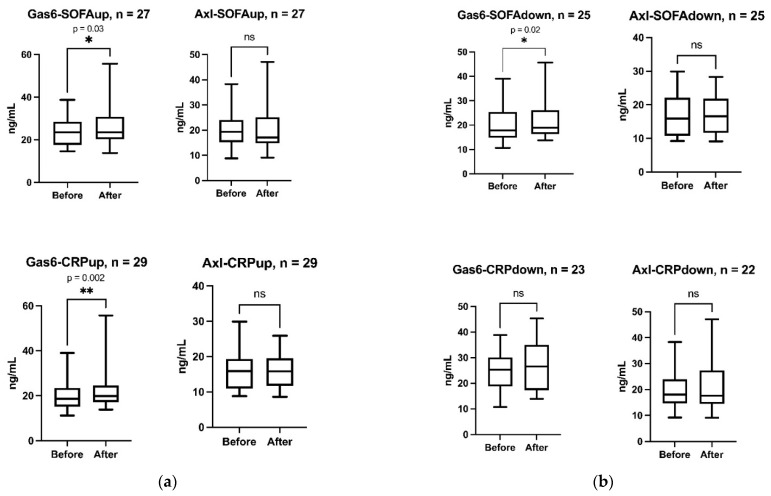
(**a**) Boxplots of Gas6 and sAxl (Axl) changes before and 24 h after vitamin K1 in patients with increased sequential organ failure assessment (SOFA; SOFAup) or C-reactive protein (CRP; CRPup). (**b**)**.** Boxplot s of Gas6 and sAxl (Axl) changes before and 24 h after vitamin K1 in patients with decreased SOFA (SOFAdown) or CRP (CRPdown). ** (*p* = 0.002), * (*p* = 0.02) and ns (no significant change).

**Figure 3 nutrients-13-04101-f003:**
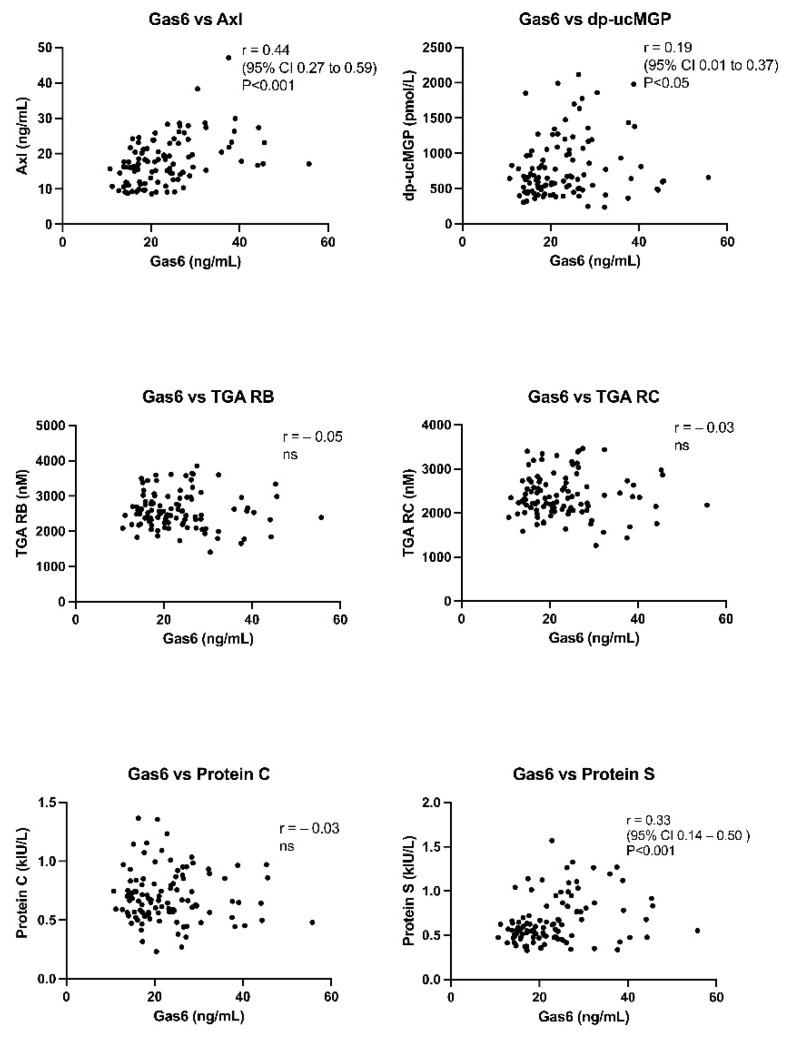
Correlations between growth arrest-specific gene 6 protein (Gas6) and soluble-Axl receptor (Axl), dephospho-uncarboxylated MGP (dp-ucMGP), thrombin generation assay (TGA), protein C and S. TGA was performed with two different reagents: TGA reagent B (RB), which has a low concentration of phospholipids and tissue factor (TF), and TGA reagent C (RC) high, which has a high concentration of TF. Ns (no significant change).

**Figure 4 nutrients-13-04101-f004:**
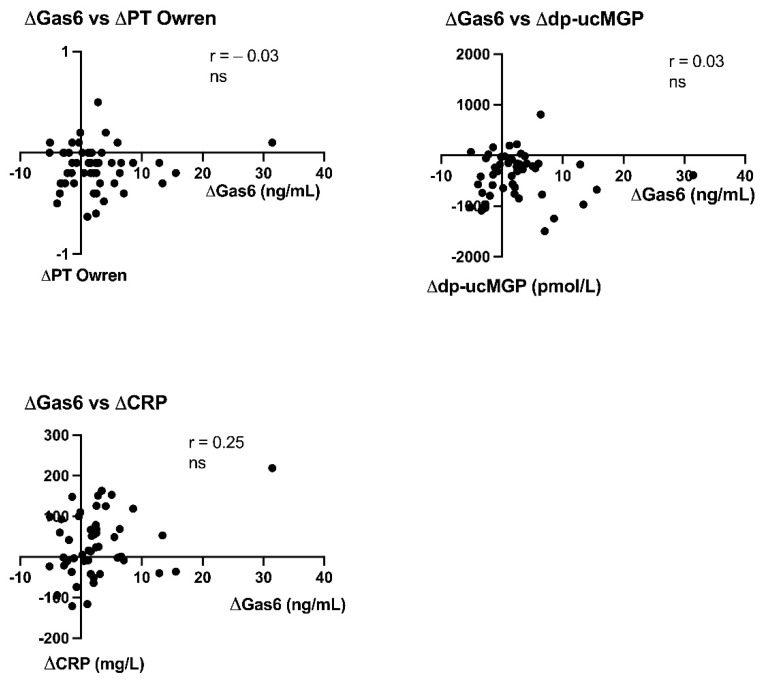
Correlations between delta-changes in growth arrest-specific gene 6 protein between sampling occasion (ΔGas6) and delta-changes in PT Owren INR (ΔPT Owren), desphospho-carboxylated MGP (Δdp-ucMGP) and C-reactive protein (ΔCRP) for the 52 patients. ns (no significant change).

## Data Availability

The datasets used and/or analyzed in the current study are available from the corresponding author on reasonable request.

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
