# Peer review of "Vitamin K Effects on Gas6 and Soluble Axl Receptors in Intensive Care Patients: An Observational Screening Study"

_nutrients, 2021, doi:10.3390/nu13114101_

Round 1

Reviewer 1 Report

Interesting manuscript with some relevant information. The work has been done previously and has been researched extensively. Perhaps this work can also be added to the literature. 

Author Response

Reviewer 1: Interesting manuscript with some relevant information. The work has been done previously and has been researched extensively. Perhaps this work can also be added to the literature. 

Response: We agree there are many publications on Gas6 and its interaction with the TAM receptors. Both blocking by warfarin and supplementation with vitamin K and K2 (often menaquinone 7) has been adressed. But there have been no report on vitamin K1 supplementation (intravenously (iv)) to critical ill patients and its effects on Gas6 with or without TAM receptor analyses. As we have identified a deficiency in another extrahepatic vitamin K dependant protein - Matrix Gla Protein (MGP) by analysing its decarboxylated glukoprotein - dp-ucMGP, that was corrected by the iv vitamin K injection in the main published study, we performed this substudy with extra analyses of Gas6 and Axl-receptor with the help of Cecilia Axelsson a previous collaborator to Prof Björn Dahlbäck, Lund University, Malmö, who has published extensively on GAs6 and Axl-receptors. You can see in methods that the analyses need high qualified laboratory technicians. The methods were controlled from healthy donor plasma. As there are concerns that  increased Gas6 can induce thromboembolism and that its antiapoptic effects can increase certain tumour growth, this is surely of importance in a postoperative or intensive care situation. This is a small observational study, but a first step, and we are already studying specific patient groups with outcome parameters and advanced laboratory monitoring (eg glukokalyx shedding markers).

Reviewer 2 Report

This article claims a correlation between vitamin K inyection and an increase in plasma concentration of protein Gas6 in 52 intensive care patients. The correlation is clear, but the presentatios of the results and the conclusions are poorly presented.

Mayor issues:

The authors analysed concentrations of several proteins in plasma, but only after 24 h after the inyection. Added to the fact that not always the samplig was at the same time after inyection (20-28 h, quite a lot), they did not evaluate the levels after more than 24 h (48-72 h). If the authors claimed the possible role of that increasing in Gas9 protein, they need to demostrated that the increasing is stable in the time. Specially due to the fact that the patients did not have a hypercoagulative response, so the role of Gas9 here is correlated with another response. The authors discussed about Gas9 and cancer, but a 24-h response is not probably related to such an illness. Besides, they indicate the the receptor Ax1 is not increased in plasma. Besides, n is too small to conclude.

More experiments are needed

Minor issues:

A list of abreviations is needed.

Lines 25-26: change the order of those sentences.

Line 46: ref 2 is not proper here.

Line 64: ref 9 is not proper here.

Line 73: ref 12 is not proper here, but you can use ref 22 in 12.

Lines 74-76: grammar is not correct.

Lines 85-86: idem

Line 105: 2014 is not recent.

Line 127: an space is lacking after K1

Results: indications of n is needed in each paragraph and figure.

Line 189: indicate what you are measuring: concentrations in plasma... before and after...

Idem in Line 213.

Line 219: results not shown, I suppose.

Fig 3: Letters are too small.

lINE 234: CI is not used at the figure

lINE 257: Indicate that you measure concentrations.

Line 261: ref 9 is not proper here.

Line 289: in vitro in italics.

Line 312: Gas9

Line 407. title repeated

Author Response

Comments and Suggestions for Authors

Lines 25-26: change the order of those sentences.

Response: done

Reviewer: This article claims a correlation between vitamin K inyection and an increase in plasma concentration of protein Gas6 in 52 intensive care patients. The correlation is clear, but the presentatios of the results and the conclusions are poorly presented.

Response: thank you for this comment. The following amendments have been done in the presentation of the results and the conclusion:
• A Graphical Abstract has been added to the submission for clarity and provides a  self-explanatory image to appear alongside with the abstract appearing on the Table of Contents
• Changes in Figure 3 and 4 to clarify
• The conclusion at the end of the Discussion is clarified and changed to: “Gas6 increased 24 hours after intravenous vitamin K1 administration, but there was no change in the level of s-Axl receptor.” 

Reviewer: Mayor issues:

The authors analysed concentrations of several proteins in plasma, but only after 24 h after the inyection. Added to the fact that not always the samplig was at the same time after inyection (20-28 h, quite a lot), they did not evaluate the levels after more than 24 h (48-72 h). If the authors claimed the possible role of that increasing in Gas9 protein, they need to demostrated that the increasing is stable in the time. Specially due to the fact that the patients did not have a hypercoagulative response, so the role of Gas9 here is correlated with another response. The authors discussed about Gas9 and cancer, but a 24-h response is not probably related to such an illness. Besides, they indicate the the receptor Ax1 is not increased in plasma. Besides, n is too small to conclude.

More experiments are needed

Response: we agree that there is no evidence on the persistence of the changes demonstrated. However, given that the current study is a sub-study of the recently published main study, we cannot add any analyses or samples. However to adress this we have added a sentence at the end of the limitation paragraph:"Furthermore, the persistence of any changes between the samples cannot be evaluated. in the current study." 

Minor issues:

A list of abreviations is needed.

Response: if the editor demands it we will comply, otherwise abbreviations have been defined when first mentioned in text.

Line 46: ref 2 is not proper here.

Response: omitted.

Line 64: ref 9 is not proper here.

Response: changed to Korshunov V. A. (2012). Axl-dependent signalling: a clinical update. Clinical science (London, England : 1979)122(8), 361–368. https://doi.org/10.1042/CS20110411

Line 73: ref 12 is not proper here, but you can use ref 22 in 12.

Response: done

Lines 74-76: grammar is not correct.

Response: sentence changed into “However, this is a complex field and other tumours can be inhibited by an increased Gas6 and TAM expression [8].”    

Lines 85-86: idem.

Response: in atherosclerotic plaques Axl is downregulated, but MerTK receptors from invasive macrophages are upregulated and interact with protein S [17].         

Line 105: 2014 is not recent.

Response: deleted.

Line 127: an space is lacking after K1

Response: added.

Results: indications of n is needed in each paragraph and figure.

Response: thank you for pointing this out. All results derives from analyses of 52 samples if not otherwise stated. We have added a sentence about this in the first paragraph of the Result section: "If not otherwise stated n=52 in all paragraphs and Figures."

added to paragraphs.  Subgroup analyses in Figure 2 differ from n=52 in Fig 1a and b. I think its unnecessary to add n=52 in these figures. The number of patients had been added to text and in the pragraphs for the rest of the figures.

Line 189: indicate what you are measuring: concentrations in plasma... before and after...

Response: changed to: Gas6 and sAxl plasma concentrations before and after vitamin K

Idem in Line 213.

Response: changed to: Boxplots of plasma concentrations of Gas6 and sAxl (Axl) before and 24h after vitamin K1 in patients with an increased Sequential Organ Failure Assessment (SOFA; SOFAup) or an increased C-reactive protein (CRP; CRPup).

Line 219: results not shown, I suppose.

Response: If its okey we rather keep “…..other analyses”

Fig 3: Letters are too small.

Response: The font has been enlarged in Figure 3 and 4

lINE 234: CI is not used at the figure

Response: deleted

lINE 257: Indicate that you measure concentrations.

Response: I do not find this paragraph, the paragraph above Line 257 (e.g. line 247) is introduced with “The main objective of this study was to investigate the potential effects of vitamin K1 treatment on Gas6 concentrations and its soluble receptor sAxl in plasma.”

Line 261: ref 9 is not proper here.

Response: I read “Protein S can bind to Axl, but with less affinity than for Gas6 [9].” But this is on line 271. I have changed reference 9 before, see above and hope this reference is adequate (Korshunov V. A. (2012). Axl-dependent signalling: a clinical update. Clinical science (London, England : 1979)122(8), 361–368. https://doi.org/10.1042/CS20110411)

Line 289: in vitro in italics.

Response: I find it on line 303, changed

Line 312: Gas9

Response: we can´t find this default

Line 407. title repeated

Response: in reference list,line 421, ref 21 - corredcted

Reviewer 3 Report

Overall a very interesting manuscript reviewing updated data from an exploratory topic previously published in Nutrients evaluating the effects of Vitamin K on Gas6 and soluble Axl receptors. Ultimately, the authors determined that there was a significant increase in Gas6 without a change in Axl after vitamin K supplementation. 

  • In the introduction you question if Vit K should be used in critically ill cancer patients due to the impact on thromboembolic events in this patient population. The authors do not comment on if any of the 52 patients in the patient population had prior malignancy. Please clarify.
  • The statistical analysis is suboptimal. Was any power analysis performed to determine that samples from 52 patients would be adequate? No power analysis is described. The groups were further divided based on SOFA score and CRP. These groups are  underpowered to determine true clinical significance. Also, the data is likely not normally distributed and further calculations other than two tailed analysis should be employed. I recommend adding a biostatistician to you co-authors and performing this analysis again. 
  • If similar conclusions are met after appropriate statistical methods are employed I think this manuscript is novel and worthy of publication. 

Author Response

Overall a very interesting manuscript reviewing updated data from an exploratory topic previously published in Nutrients evaluating the effects of Vitamin K on Gas6 and soluble Axl receptors. Ultimately, the authors determined that there was a significant increase in Gas6 without a change in Axl after vitamin K supplementation. 

  • In the introduction you question if Vit K should be used in critically ill cancer patients due to the impact on thromboembolic events in this patient population. The authors do not comment on if any of the 52 patients in the patient population had prior malignancy. Please clarify.

Response: This is a good point. Although many of the baseline variables for the investigated cohort were presented in the original publication (ref 26 in the current manuscript), we have now added information on priori malignancy also in the first paragraph of the Result section in the current manuscript: The studied cohort has previously been described [21]. In summary 69 % were male, the median age was 68 years (range: 55–74 years), septic shock was the most common diagnosis (29%) followed by cardiovascular disease (13%) Three patients (6%) were diagnosed with cancer.   

  • The statistical analysis is suboptimal. Was any power analysis performed to determine that samples from 52 patients would be adequate? No power analysis is described. The groups were further divided based on SOFA score and CRP. These groups are  underpowered to determine true clinical significance. Also, the data is likely not normally distributed and further calculations other than two tailed analysis should be employed. I recommend adding a biostatistician to you co-authors and performing this analysis again. 

Response: The power analysis was presented in the original publication (ref 26 in the current manuscript) and was based on the effect size of the primary outcome in that study:  Sample size calculation was performed using G*Power version 3.1 (Heinrich Heine Universität, Düsseldorf, Germany) for the Wilcoxon signed-rank test and was based on a pilot study from our department where 10 mg Konakion Novum® was given intravenously to non-bleeding critically ill patients with prolonged Owren PT. The effect size in that study was 0.48 and was based on changes in Owren PT. With a two-tailed α-value of 0.05 and 90% power, the target sample size was set to 50 patients for this study.

To further clarify this we have added text to describe this shortcoming in the limitation paragraph of the Discussion section: Furthermore, the study is explorative, without sample size calculation for any outcomes and hypothesis-generating...

We agree that the power of the sensitivity analyses are smaller than in the main analyses and that this increases the risk for a Type 2 statistical error. This is further clarified in the 3.3 paragraph (Subgroup analyses) of the Result section: . It should be noted that the number of observations are lower than in the main analyses which increases the risk for statistical Type 2 errors. 

We also agree that none of the variables were normally distributed and that non-parametric tests are the appropriate tests to use. Therefore, this is already described in the 2.5. Statistical analyses description of the Methods section: ...was utilised to do a two-tailed Wilcoxon matched-pairs signed rank test that was in turn used to estimate changes in the before and after levels of Gas6 and sAxl receptor. Spearman correlation analyses were used to evaluate correlations. All variables were found to have non-Gaussian distributions.

  • If similar conclusions are met after appropriate statistical methods are employed I think this manuscript is novel and worthy of publication. 

Round 2

Reviewer 2 Report

My main opiniosn is the same, the conclusions are not supported boy the results.

Added to the fact that not always the samplig was at the same time after inyection (20-28 h, quite a lot), they did not evaluate the levels after more than 24 h (48-72 h). If the authors claimed the possible role of that increasing in Gas9 protein, they need to demostrated that the increasing is stable in the time. Specially due to the fact that the patients did not have a hypercoagulative response, so the role of Gas9 here is correlated with another response. The authors discussed about Gas9 and cancer, but a 24-h response is not probably related to such an illness. Besides, they indicate the the receptor Ax1 is not increased in plasma

Author Response

We have modified the message in the revised draft, looked at singuylar responses in Gas6 and Axl from Figures 1,3 and 4 yo better understand our results. The small increase in Gas6 noted in Figure 1 is probably not related to a vitamin K effect - we agree with the reviewer. To bring on this research vitamin K effects on the carboxylationof Gas6 needs to be adressed. We will proceed to develop such an ELISA test for humans.